# Has-miR-129-5p’s Involvement in Different Disorders, from Digestive Cancer to Neurodegenerative Diseases

**DOI:** 10.3390/biomedicines11072058

**Published:** 2023-07-21

**Authors:** Adrian Boicean, Sabrina Birsan, Cristian Ichim, Ioana Boeras, Iulian Roman-Filip, Grama Blanca, Ciprian Bacila, Radu Sorin Fleaca, Horatiu Dura, Corina Roman-Filip

**Affiliations:** 1Faculty of Medicine, Lucian Blaga University of Sibiu, 550169 Sibiu, Romania; adrian.boicean@ulbsibiu.ro (A.B.); cristian.ichim@ulbsibiu.ro (C.I.);; 2Molecular Biology Laboratory of the Applied Ecology Research Center, Faculty of Sciences, Lucian Blaga University of Sibiu, 550012 Sibiu, Romania; ioana.boeras@ulbsibiu.ro; 3Department of Neurology, “George Emil Palade” University of Medicine, Pharmacy, Sciences and Technology, 540136 Targu Mures, Romania; 4Faculty of Social Sciences, Lucian Blaga University of Sibiu, 550012 Sibiu, Romania

**Keywords:** miR-129-5p, downregulation, digestive neoplasia, neurological diseases

## Abstract

At present, it is necessary to identify specific biochemical, molecular, and genetic markers that can reliably aid in screening digestive cancer and correlate with the degree of disease development. Has-miR-129-5p is a small, non-coding molecule of RNA, circulating in plasma, gastric juice, and other biological fluids; it plays a protective role in tumoral growth, metastasis, etc. Furthermore, it is involved in various diseases, from the development of digestive cancer in cases of downregulation to neurodegenerative diseases and depression. Methods: We examined meta-analyses, research, and studies related to miR-129-5-p involved in digestive cancer and its implications in cancer processes, as well as metastasis, and described its implications in neurological diseases. Conclusions: Our review outlines that miR-129-5p is a significant controller of different pathways, genes, and proteins and influences different diseases. Some important pathways include the WNT and PI3K/AKT/mTOR pathways; their dysregulation results in digestive neoplasia and neurodegenerative diseases.

## 1. Introduction

In recent years, teams of researchers have intensively studied the involvement of RNA molecules (specifically, miRNAs) in the digestive oncogenetic process. Cellular apoptosis has been shown to have tissue specificity and miRNAs can be considered true biomarkers of the tumor process [1,2,3]. miRNAs have significant importance in pathology, due to their stress response and modified expression throughout disease evolution and progression. Considerable attention has been paid to clarifying the roles and alterations that miRNAs suffer in the development and progression of cancer, possibly leading to important clinical and therapeutical progress in molecular research. Furthermore, in vitro studies note miRNAs as promising candidates in molecular replacement therapy in order to halt cancer progression and lymph node metastasis, and to induce the apoptosis of tumoral cells. Also, each study on miRNAs brings us closer to a molecular vision of oncogenesis and neurodegenerative diseases. Recent studies have outlined the importance of studying miRNAs and the genes they target for their potential roles in the personalized treatment of various diseases [1,2,4].

To date, more than 2500 miRNAs have been considered as crucial regulators in tumor proliferation, growth, and metastasis; these molecules could pave the way for the targeted therapeutic management of patients [5,6,7,8,9]. Considering the fact that circulating miRNAs represent important regulators in intercellular communication and changes in tissue expression, as well as their stability in both tissue and biological fluids, their clinical potential should be emphasized. MicroRNAs represent non-coding molecules that are formed from 21 to 23 nucleotides. Due to the fact that they circulate in body fluids, recent research outlines their potential as disease biomarkers that can be used for screening as well as for disease progression and therapeutical prognosis [5,6,7]. We note the importance of miRNAs in the development of many diseases; a single miRNA can regulate a variety of genes, and a single gene can be targeted by many miRNAs. This is why miRNAs are involved in development, proliferation, apoptosis, metabolism, differentiation, and metastasis [6,7,8].

According to a meta-analysis conducted by Diana Bautista-Sánchez et al., miRNA functions have implications in the homeostatic regulation of gene expression as well as in cellular responses [7]. Homeostatic regulation of gene expression refers to regulation of genes according to cell cycles and their requirements. The cellular response is defined by cellular differentiation, the cell’s lifecycle, and its response to stress. The importance of using miRNAs as biomarkers is researched and studied due to their molecular stability in tissues as well as in fluids, which makes them easy to use and evaluate from body fluids like plasma, ascites, and gastric juice [7,8,9].

Recent research is increasingly aimed at exploiting and capitalizing on the potential of miRNAs, in either plasma/serum or other biological fluids, to establish non-invasive screening protocols that can detect early-stage digestive neoplasia or other pathologies [6,7].

Digestive cancers represent an important issue for gastroenterologists, surgeons, and oncologists in terms of early detection and therapeutical management, as they represent some of the most common types of cancers in the world. In this review, we consider neoplasias of the digestive tract, including stomach and colorectal neoplasias, and cancers affecting other digestive organs like the pancreas and liver. The etiology and pathogenesis of digestive cancers are multifactorial. Recent research has focused increasingly on molecular changes in tumoral tissues in order to obtain early diagnoses and the possibility of targeted treatment to lower adverse effects [7,8,9,10].

The International Agency for Research on Cancer estimates that, on average, 3.5 million GI cancer deaths occur worldwide each year, which emphasizes the need for in situ diagnostics and improvements in oncological treatment. Studies on miRNAs give hope for early diagnoses in neoplasia as well as other diseases due to the pathways they signal in addition to their clinical benefits. Another important benefit of miRNAs is that they can be extracted from blood, serum, gastric acid, bile, and other biological fluids; further, they preserve their specificity and sensibility as biomarkers [8,9,10,11,12,13].

A specific miRNA can regulate, by post-transcriptional mechanisms, up to about 60% of protein-coding genes. Furthermore, miRNAs are implicated in cell growth, homeostasis, apoptosis, and cell migration. In brain development, a single miRNA may be implicated in many processes, from synaptic formation to neural development. Current molecular research focuses on clinical modifications in cases of miRNA dysregulation in order to establish targeted treatment in different diseases [6,14,15,16].

Hsa-miR-129-5p plays a crucial role in suppressing tumor growth, proliferation, and metastasis. Moreover, a low level of has-miR-129-5p is statistically correlated with more aggressive forms of cancer and is related to the appearance of lymph node metastases and peritoneal carcinomatosis. A low level of miR-129-5p is associated with resistance to chemotherapy, and studies show that manipulating the level of microRNA 129 in vitro can improve the response to chemotherapy and prolong survival [14,15,16,17,18,19,20,21,22,23]. 

The principal pathways to which miR-129-5p is related, according to studies, are WNT/β-catenin and PI3K/AKT/mTOR. MiR-129-5p targets the WNT signaling/B-catenin pathway, which plays a role in cell adhesion and tumor-cell-line proliferation in the case of dysregulation by affecting cadherin/catenin ligands and resulting in the proliferation of gastric cancer. Downregulation of miR-129-5p results in the dysregulation of WNT pathways because, in normal tissues, miR-129-5p acts as an inhibitor for several components of the WNT signaling pathway. WNT/β-catenin is the most-studied pathway involved in controlling the proto-oncogene β-catenin, which, being a multifactorial protein, is also part of the E-cadherin-binding protein implicated in the malignant process [10,12]. In the same vein, studies show that inappropriate signaling of the WNT/β-catenin pathway results in the development of neurodegenerative diseases such as Alzheimer’s or Parkinson’s disease [12].

These are some of the most important signaling pathways involved in regulating the cell cycle (proliferation, invasion, migration) of tumoral cells, and they are dysregulated in various types of digestive cancer, as well as in neurodegenerative diseases. Furthermore, through the abnormal regulation of phosphoinositide 3-kinase (PI3K)/protein kinase B (Akt), which targets the mTOR pathway, this type of miRNA influences the development of neurological diseases; downregulation of has-miR-129-5p results in neuroinflammation and the development of neurodegenerative diseases, whereas upregulation results in protection from neural apoptosis [14,15,16].

MiR-129-5p also targets ADAM proteins, playing an important role in cancer development and progression, options for treatment, and drug resistance. Furthermore, the inhibition of high-mobility group protein B1 (HMGB1) is a target of miR-129-5p and, according to some studies, results in cancer development in cases of dysregulation [22,23]. This relationship between the types of microRNAs that target various proteins involved in cancer genesis outlines the importance of understanding the molecular mechanism of carcinogenesis and emphasizes the role of microRNAs as biomarkers in screening early cancers [17,20,23]. 

In the same vein, we note the fact that miR-129-5p targets similar pathways (PI3K/AKT/mTOR pathways) in both digestive diseases and in neurodegenerative diseases. Long HZ et al. noted that signaling the PI3K/AKT pathway in neurodegenerative diseases prevents neurotoxicity and improves neuron survival in Alzheimer’s disease (AD) and Parkinson’s disease (PD) [4]. Based on the same signaling pathways, like PI3K/AKT/mTOR (targeted by miR-129-5p), experimental studies are trying to improve the therapeutic means based on RNA that could lead to personalized, targeted, therapeutic management, with better specificity than classical treatments for digestive cancers and neurodegenerative diseases. Future perspectives that involve miRNA replacement therapy could lead to a network of tumor-suppressive clinical benefits in various diseases that involve the downregulation of miR-129-5p [24,25,26,27].

In this review, we emphasize the importance of the molecular dysregulation of miR-129-5p in the development and progression of gastrointestinal cancer, as well as in neurodegenerative diseases.

To our knowledge, this is the first review that describes the molecular changes that occur in cases of downregulation of this type of miRNA from the gastrointestinal and neurological perspectives. We also outline the importance of developing molecular biomarkers that could predict the possible development of different diseases. Furthermore, we highlight that through development of the different pathways that miR-129-5p signals, miR-129-5p has clinical importance from tumoral cells, lymph-node metastasis, and proliferation to synaptic formation and neural apoptosis.

## 2. Methods

The review was realized according to the Preferred Reporting Items for Systematic Reviews and Meta-Analysis Guidelines 2020 (PRISMA), based on inclusion and exclusion criteria. As search engines, we used Google Scholar, Web of Science, PubMed, and Up-to-Date. In the period 2013–2023, over 1000 specialized studies were published with reference to the implications of miRNAs in oncogenesis, but very few were published concerning microRNAs in neurodegenerative diseases. We chose to study the most recent analyses and research related to miR-129-5p being involved in digestive cancer and neurological disorders. We considered systematic reviews, meta-analyses, and original studies on miR-129-5p and its implications in cancer processes and metastasis, and also analyzed its implications in neurological diseases. Keywords used for searching the databases were “microRNA”, “microRN129-5-p”, “gastric cancer”, “colorectal cancer”, “Parkinson’s diseases”, “Alzheimer’s diseases”, “the Fragile X Mental Retardation gene 1 *(FMR1*)”, and others related to miR-129-5p. 

The inclusion criteria for studies were peer-reviewed journals, studies, and meta-analyses regarding the involvement of miR-129-5p in digestive cancer and neurodegenerative diseases, and the implications of different signaling pathways and targeted proteins. Exclusion criteria were as follows: not the subject of a peer review, single case reports, studies related to other miRNAs besides miR129-5p, and other types of cancer like lung cancer or glioblastoma related to miR-129-5p.

The primary outcome was to assess the implications of miR-129-5p in various diseases like digestive neoplasia and neurological diseases. We also described the common pathways between these pathologies. The secondary outcome was to outline studies that presented potential therapeutic perspectives for this type of miRNA.

## 3. Results and Discussion

The database search identified 474 records, including 175 duplicates. A total of 223 articles were selected for screening; 100 of them were excluded and another 123 were sought for retrieval. A total of 68 articles were evaluated for eligibility, 55 of which were included in the final analysis; 37 were original reports (we included human studies as well as animal models) and 18 were reviews (Figure 1).

### 3.1. Studies on miRNA 129 in Digestive Cancer

Although several miRNAs with altered expression have been studied in cases of digestive cancer, we focused on miR-129-5p and its clinical implications in different types of digestive cancer [28,29,30].

One of the most lethal cancers involving the digestive tract and linked to the dysregulation of miR-129-5p, is pancreatic cancer. Other studies on miR-129-5p show its importance in the proliferation and development of gastric and colorectal cancers. A study conducted by Zhisheng Qiu et al. noted that overexpression of this type of miRNA suppressed the proliferation of tumoral pancreatic cells. Also, miR-129-5p played a role in suppressing cell proliferation and invasion by regulating the *PBX3* gene and induced apoptosis in tumoral cells. The same study noted that the suppression of miR-129-5p had the opposite result by promoting the development of pancreatic cancer in cases of under-expression. Their study involved 30 patients with pancreatic cancer (PC) and concluded that the level of miRNA in tumoral cells was statistically correlated with metastasis and clinical stages. It was also reported that the level of miRNA was correlated with the survival of the patients, which is an important clinical prediction tool for practitioners [28].

Another study that outlined the connection between the downregulation of miR-129-5p and pancreatic cancer was by Jin Xu et al., who analyzed 55 samples of pancreatic cancer tissue vs. adjacent healthy tissue. They noted that miR-129-5p is a posttranscriptional controller for the IPO7 protein and its suppression resulted in IPO7 upregulation in pancreatic tumoral tissue and cells. Furthermore, they concluded that upregulation of IPO7 reduces the expression of *P53* gene, resulting in the suppression of miR-129-5p and leading to positive feedback that promotes the development of pancreatic cancer. Since each work on molecular oncogenesis helps to clarify the cellular mechanism of cancer in different pathologies, there is hope that in the future this may lead to cancer prevention and targeted treatments [1].

The same type of miRNA is involved in hepatocarcinoma and also interferes with chemoresistance. According to Huge et al., miR-129-5p plays a beneficial role by signaling through the WNT pathway, which causes protective regulation in hepatocellular carcinoma through downregulating hepatoma-derived growth factor HDGF. The study involved an animal model and miR-129-5p was instilled in HCC cells. The results showed the dependence of the overexpression of miR-129-5p and the clinical benefits in the suppression of hepatocellular carcinoma by directly regulating hepatoma-derived growth factor (HDGF). We highlight the importance of this study as it shows the possible therapeutical effect of miR-129-5p by regulating the WNT signaling pathway. MiR-129-5p may represent a promising clinical tool for miRNA replacement therapy in many types of cancer, especially HCC [9,10,11].

Molecular and clinical studies are focusing increasingly on understanding the action of miR-129-5p dysregulation in promoting different types of digestive cancer. Although further research is needed, currently we note that the overexpression of miR-129-5p has a protective role in digestive cancer. Z Jiang et al. observed the expression of miR-129-5p to be suppressed in gastric cancer. In their research, miR-129-5p was extracted and quantified via RT-PCR from tumoral tissue as well as from blood samples. They noted statistically significant correlation between the control group and the study group. Moreover, using a luciferase reporter assay, they outlined that IL-8 is controlled and targeted by miR-129-5p. Furthermore, IL-8 is downregulated by miR-129-5p, playing a protective role in cases of overstimulation [12]. 

Another gene targeted by miR-129-5p is *COL1A1*. Studies show that co-transfection of miR129-5-p and downregulation of *COL1A1* result in a clinical reduction in tumors and metastasis [11]. These results emphasize the important clinical benefits of this type of miRN in the targeted treatment of gastric cancer. The clear message from the above studies that measured the expression of miR-129-5p is that this type of miRNA influences the development of digestive cancer by targeting many genes through different pathways [11,12,13]. This conclusion outlines the clinical importance of miR-129-5p for practitioners and possible targeted treatments. Moreover, the results presented by Wang et al. show that the co-transfection of miR-129-5p mimics resulted in tumor suppression and prevented further proliferation and invasion, highlighting the protective role of this miRNA in cases of gastric cancer [11].

A related research direction involves considering the connection between families of transmembrane metalloproteinases, integrins, and metalloproteinase 9 (ADAM proteins), which seem to be connected with cancer genesis and metastasis, as well as chronic inflammation. Studies focus on the importance of ADAM 8, 9, and 12 and gastrointestinal cancer. Recent research has outlined that miR129-5-p plays a role in gastric cancer suppression by targeting ADAM 9. The study focused on gastric tumoral tissues from 50 patients and compared them with normal adjacent tissues. It was noted that miR129-5-p was downregulated in gastric cancer tissue and tumoral cells compared with the control samples. Furthermore, it was confirmed that miR129-5-p regulated ADAM 9, reducing proliferation and tumor growth. Also, it was noted that in vivo, the overexpression (upregulation) of this type of miRNA resulted in inhibition of tumoral growth, induced apoptosis, and prevented metastasis. ADAM 9 seems to be distinctly overexpressed when gene expression profiling is realized in cases of hepatocellular adenocarcinoma. Studies show that immunotherapy decreased the level of ADAM 9 in HCC. ADAM 9 levels are high in other types of digestive cancers like pancreatic ductal adenocarcinoma (PDAC). On the basis of these findings, we may speculate that inducing normal levels or overexpression of miR129-5-p by targeting ADAM 9 may have a protective role in many types of digestive cancers, from the gastrointestinal tract to other digestive organs like the liver or the pancreas [18,19,20,21].

Another mechanism in suppressing gastric cancer is inhibiting high-mobility group protein B1 (HMGB1). Studies found that overexpression of miR129 promoted apoptosis of tumoral cells by targeting HMGB1 (high mobility group protein B 1) in cases of gastric cancer. HMGB promotes inflammation and seems to be overexpressed in gastric cancer. This protein is usually suppressed by a normal level of miRNA. A study on 25 tumoral tissues and 25 normal tissues concluded that the expression of HMGB1 was upregulated in gastric cancer samples compared with normal control samples (*p* < 0.05). In the same study, it was outlined that the expression level of miR-129-5p in gastric cancer tissue was significantly suppressed compared with the control samples (*p* < 0.001). HMGB1 is also upregulated in other types of cancer related to the dysregulation of miR-129-5p, e.g., colorectal cancer, liver cancer, and pancreatic cancer. HMGB1 inhibits apoptosis and promotes tumoral growth and proliferation. In light of these findings, we emphasize the importance of miR-129-5p overexpression in targeting and inhibiting HMGB1 and therefore preventing tumoral growth in many digestive types of cancer (Figure 2) [22,23].

A study conducted by Chen Di et al. notes the importance of glucose metabolism in tumoral cells in cases of gastric cancer through PI3K-Akt pathways by targeting the *SLC2A3* gene. The authors noted that by regulating the glucose metabolism of gastric tumoral cells, the miR-129-5p/*SLC2A3* axis may represent a perspective in gastric cancer therapy by promoting the overstimulation of miR-129-5p and inhibiting the glucose metabolism of gastric tumoral cells, as well as proliferation and tumoral growth. As stated by the authors, glucose metabolism represents a potential molecular treatment target of gastric cancer [24].

MiR-129-5-p deregulation participates in different aspects of digestive cancer pathogenesisand its protective role in cases of overstimulation has been noted in studies of all types of digestive cancer. We emphasize the importance of further studying this type of microRNA as a potential clinical biomarker for early diagnosis or for establishing disease prognosis. Furthermore, we note that overstimulation in animal model studies resulted in a potential treatment target in cases of oncological diseases. In conclusion, we highlight the importance of up-streaming the PI3K-Akt signaling pathway with a protective role for stopping the neoplastic proliferation in digestive neoplasia. Furthermore, this pathway has clinical implications in neurodegenerative diseases.

### 3.2. Studies on microRNA 129-5-p in Neurological Diseases and Depression

Studies note that miRNAs play a crucial role in neurological diseases by controlling the inflammatory response and nerve injury. This brings new perspectives in understanding degenerative diseases and neural injury. For example, recent studies have focused on the destructive characteristics of Alzheimer’s disease (AD), which represents the most significant cause of dementia and manifests as a neurodegenerative disease that is defined by aggregates of beta amyloid (Aβ) that induce neuroinflammation and brain damage. Currently, the molecular modifications in AD are still unknown and recent studies have tried to establish the importance of miRNAs and their implications in the hyperphosphorylation of tau, which results in neurofibrillary tangles (NFTs) and amyloid plaques produced by amyloid β (Aβ) aggregation [4,30,31,32]. Due to the fact that the incidence of neurodegenerative disease is growing, new biomarkers for early diagnosis and new therapeutical perspectives are needed. MiRNAs could help to elucidate molecular changes in cases of neurodegenerative disorders, and further studies are required in order to establish promising biomarkers for early diagnosis combined with the study of the pathways involved and targeted genes [33,34,35,36].

Recent studies have indicated that upregulated miR-129-5p helps the revascularization of induced intracerebral hemorrhage in rats. Other studies note that miR-129-5p could prevent the progression of Alzheimer’s disease by suppressing neural apoptosis in rats. In degenerative neurological diseases like Alzheimer’s, miR-129-5p reduces the inflammatory neural reaction by targeting the *SOX6* gene. MiR-129-5p overexpression results in the downregulation of *SOX6*, which results in reduced neural inflammation, the prevention of neural degeneration, and a reduction in inflammatory injury [36,37]. Another study direction shows that miR-129-5p targets the PI3K/AKT/mTOR signaling pathways and, as a result, plays an important role in neurodegenerative diseases by targeting the mTOR pathway, which influences cellular migration, proliferation, and apoptosis. The molecular changes that influence this pathway play an important role in different disorders like cancer, Crohn’s disease, and many other pathologies. mTOR signaling is responsible for physiological neurodegeneration. Furthermore, the mTOR pathway is involved in neural response and oxidative stress. There also seems to be a connection between mTOR signaling and neural responses in the gastrointestinal tract. In light of this association, we note the importance of the dysregulation of has-miR-129-5p, which influences the development of gastrointestinal cancers and neurological diseases [38,39,40].

Based on the PI3K/AKT/mTOR signaling pathway, studies have developed different products that could result in upstreaming this signaling pathway. By overexpressing miR-129-5p, there is an overstimulation of PI3K/AKT/mTOR, which could prevent and treat AD and PD [4.38,39,40]. Furthermore, studies show a close relationship between the PI3K/AKT/mTOR and WNT/β-catenin pathways, as they are considered to act alike and reciprocally regulate each other. Moreover, many studies note that these pathways could be considered a single target [12]. The PI3K/AKT/mTOR and WNT/β-catenin pathways represent important signaling targets of miR-129-5p that are involved in cell regeneration, cognitive memory, and neural apoptosis; these could result in new therapeutical perspectives in a wide spectrum of diseases [12,13,14,15].

A study realized by Valerija Dobricic et al. showed the dysregulation of miRNA in neurodegenerative pathologies. Their study is one of the largest Parkinson’s/Alzheimer’s disease studies on microRNAs (n = 451), including has miR-129-5p. They found that has-miR-129-5p is significantly downregulated in Parkinson’s disease (*p* = 0.0379). Their research also includes a meta-analysis of the implications of has-miR-129-5p in Alzheimer’s disease and notes, for the first time, its implications in Parkinson’s disease. They concluded that miR-129-5p has a protective role by preventing the accumulation of amyloid B-peptide. Furthermore, they stated that in an animal study on Alzheimer’s disease, this type of miRNA reduced neural apoptosis. The implications of has-miR-129-5p in both neurological diseases could sustain the hypothesis that both pathologies share similar molecular transformations. The authors of the study also described the neuroprotective role of has-miR-129-5p in preventing neural death and degeneration. The additional results of the study suggest that hsa-miR-132-3p is associated with staging in Parkinson’s disease, whereas miR-129-5p could be considered only a biomarker in diagnosis but its dysregulation is not associated with staging in PD [40].

Previous studies have described the implications of miR-129-5p at the synapse and its implications in regulating the metabolism of *FMR1*(Fragile X Mental Retardation gene 1). Chao Wu et al. realized in situ hybridization (ISH) in a mouse brain and concluded that miR-129-5p targets *FMR1*, which is a mutant gene encountered in the autism spectrum as well as in Fragile X Syndrome. Also, in their animal model study, they emphasized that miR-129-5p could restore healthy neural development by targeting *FMR1*. They also noted that abnormal regulation of this type of miRNA influences neuronal migration by affecting multipolar to bipolar transition. They concluded that *FMR1* is a direct target of miR-129-5p but not miR-129-3p. Although the study was performed using animal models, it provides perspectives and knowledge with regard to the importance of miR-129-5p in cortex development, neural migration, and its relationship to other genes and proteins that are implicated in the autism spectrum and Fragile X Syndrome. In the same vein, Zongaro S et al.’s animal model outlined that overexpression of miR-129-5p resulted in reducing the expression levels of the Fragile X Mental Retardation gene 1 (*FMR1*) by almost 40%. These studies present similar results concerning the miR-129-5p target of *FMR1* and its positive results in cases of overexpression, and note the importance of miR-129-5p in processes from synaptic formation to neural apoptosis [41,42,43,44,45,46,47] ( Table 1.)

Currently, this type of miRNA has clinical implications both in carcinogenesis, as in neurodegenerative disease, or other neural development pathologies like Fragile X Syndrome caused by the deregulation of *FMR1*, a target of miR-129-5p. MiRNAs are molecules that respond to stress; in this context, we note that physiological stress causes dysregulation in the expression of miRNAs, showing a higher prevalence in the central amygdala as well as in the hippocampus. In addition, chronic stress results in dysregulation of miRNAs in the frontal cortex, causing depression and neurodegenerative diseases [46,47,48,49,50,51,52]. A study realized by Qinlin. et al. in an animal model concluded that miR-129-5p overexpression could reduce depression in mice by targeting FEZ1/SCOC/ULK1/NBR1 proteins (*p* < 0.05), therefore outlining the importance of miR-129-5p in depression [53,54,55,56,57,58,59]. In conclusion, we note the importance of miR-129-5p in targeting different pathways and interacting with different genes. Furthermore, we outline that studies note the potential therapeutical benefits of overstimulating miR-129-5p in preventing neurodegenerative diseases and depression (Figure 3.)

## 4. Conclusions

Our review has shown that miR-129-5p is a crucial controller of various pathways involved in different diseases. Some important pathways are WNT and PI3K/AKT/mTOR, and their dysregulation results in digestive neoplasia and neurodegenerative diseases. Also, miR-129-5p has been implicated in the regulation of ADAM 9 and HMGB1, and the over-expression of miR-129-5p offers protection towards cell proliferation and metastasis in cases of digestive cancer. We also note the implication of acute and chronic stress in the dysregulation of this type of miRNA, resulting in neuroinflammation, neural apoptosis, and the development of depression. Considering that acute and chronic stress may be causes of the dysregulation of miR-129-5p, we note the relationship between molecular regulation in the brain and in the digestive tract. We can conclude that downregulation of this microRNA as a result of stress may suppress the inhibition of high-mobility group protein B1 (HMGB1) and result in tumor proliferation.

We emphasize that the overexpression of miR-129-5p interferes with the pathways of several targeted genes and has protective and therapeutic potential for many diseases. Furthermore, although miR-129-5p has been studied and measured in different types of gastrointestinal cancers, it also presents an important marker in brain development and neurodegenerative diseases, depression, and responses to chronic and acute stress.

To conclude, we outlined the clinical importance of miR-129-5p as a screening non-invasive biomarker related to various diseases due to the pathways, genes, and proteins it regulates. Furthermore, replacement therapy studies have noted a potential clinical benefit from controlling signaling pathways, which could represent a promising treatment strategy in various diseases related to the dysregulation of miR-129-5p. Further studies on this type of miRNA will provide new therapeutical opportunities for practitioners in oncology, neurology, and other medical fields.

## Figures and Tables

**Figure 1 biomedicines-11-02058-f001:**
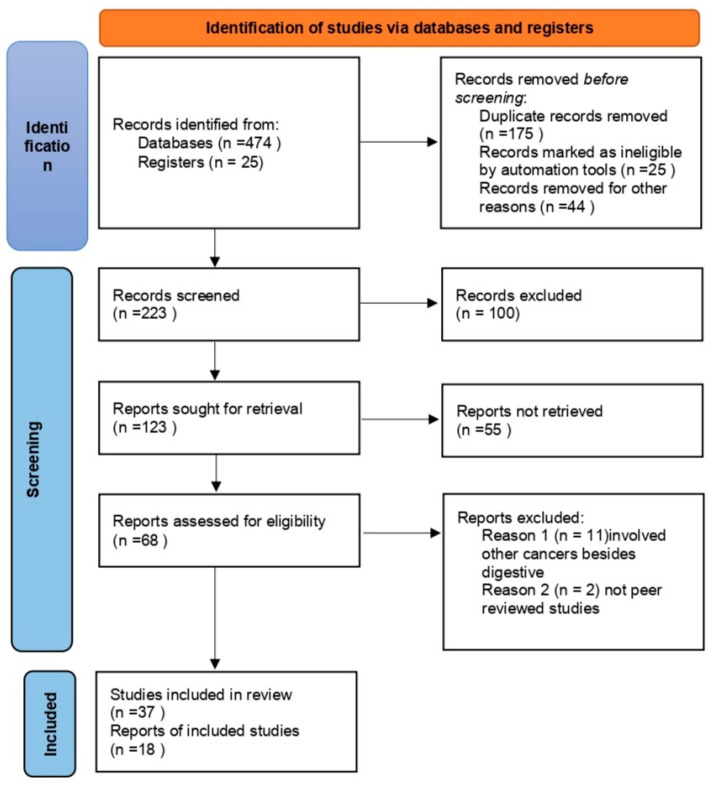
Research strategy.

**Figure 2 biomedicines-11-02058-f002:**
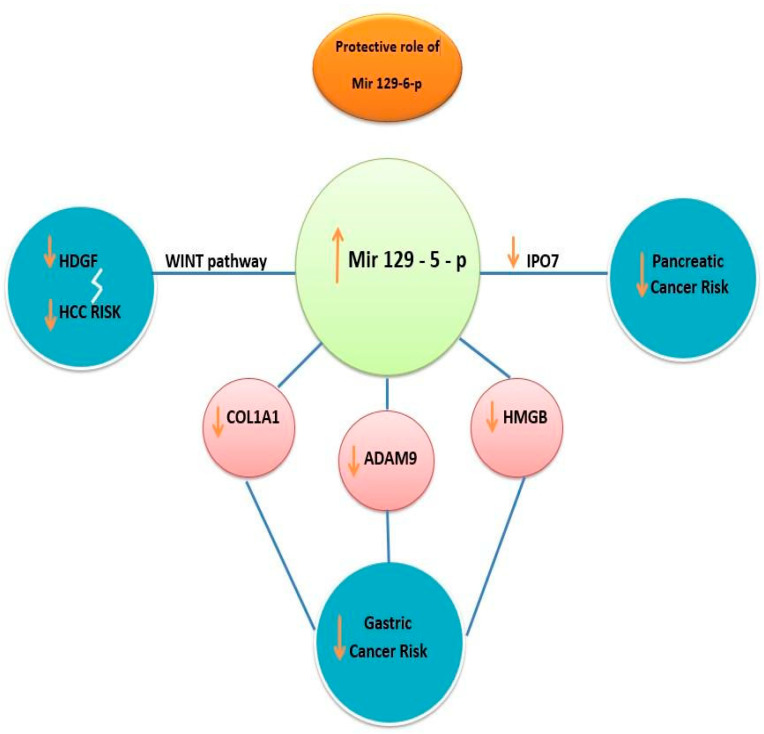
Protective role of miR-129-5p in preventing tumoral progression in different types of cancer. Overstimulation of miR129 targets WNT signaling pathway resulting in decreasing hepatoma-derived growth factor (HDGF) and HCC risk. Upregulation of miR-129-5p decreases *COL1A1*, ADAM 9 protein, and HMGB protein, lowering gastric cancer risk and decreasing IPO7 protein, resulting in suppressing pancreatic cancer proliferation.

**Figure 3 biomedicines-11-02058-f003:**
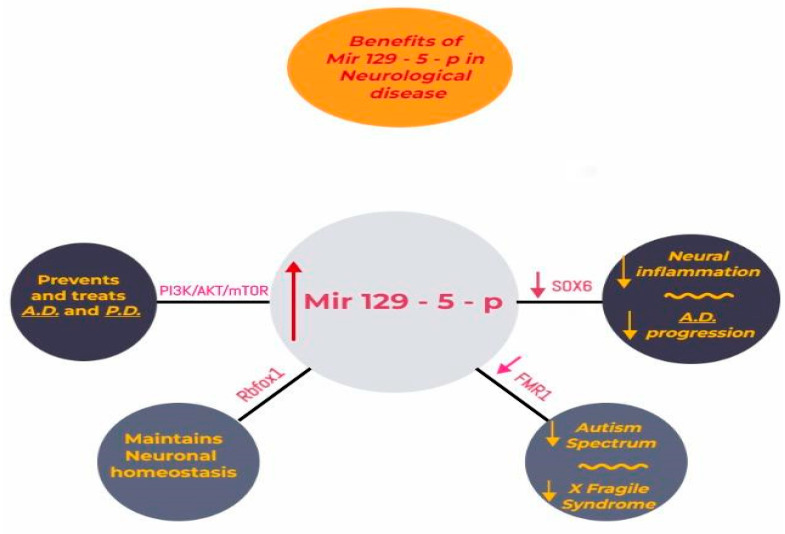
Overstimulation of MiR-129-5p plays a protective role in reducing neural inflammation and preventing Alzheimer’s disease (AD) by reducing the expression of the *SOX6* gene. Controlling the signaling pathway PI3K/AKT/mTOR represents a possible treatment in AD and PD. Upregulation of miR-129-5p targets genes and maintains neuronal homeostasis. Also, by lowering the activity of the mutant *FMR1* gene, miR-129-5p reduces the risk for diseases in the autism spectrum and Fragile X Syndrome.

**Table 1 biomedicines-11-02058-t001:** HSA-miR- 129-5p dysregulation and its clinical implications.

Study	Results	Additional Results/Details
Pancreatic cancer		
Zhisheng et al. [28]	- 30 patients evaluated with pancreatic cancer. - Overexpression of miR-129-5p suppressed proliferation and migration of pancreatic tumor and promoted apoptosis in tumoral cell lines. - Suppression of miR-129-5p induced pancreatic tumoral cell development.	Level of miR-129-5p is statistically correlated with the rate of metastasis, clinical stage, and patient survival.
Jin Xu et al. [1]	- 55 patients evaluated with pancreatic cancer. - Inhibition of miR-129-5p led to IPO7 overexpression, promoting pancreatic cancer development.	miR-129-5p and IPO7 provide valuable insights and indications for the diagnosis and treatment of pancreatic cancer.
Other cancer types		
Huge et al. [10]	- Mouse model miR-129-5p transfection of HCC cells. - miR-129-5p downregulates HDGF with HCC tumor suppressive effect.	HDGF dysregulation is linked with poor survival in patients with adenocarcinoma. Also, HDGF expression is correlated with WNT pathway dysregulation, targeted by miR-129-5p.
Jiang et al. [12]	- 50 patients with gastric cancer. - miR-129-5p was downregulated in GC patients. - miR-129-5p also suppressed cell proliferation and invasion.	miR-129-5p exerted its effect through the inhibition of interleukin8.
Neurological diseases		
Dobricic et al. [40]	- 458 patients with Parkinson’s/Alzheimer’s disease. - Has-miR-129 -5p is downregulated in Parkinson’s disease. - miR-129-5p determined neuroprotective roles against amyloid β-peptide accumulation.	Hsa-miR-132-3p is associated with α-synuclein Braak staging in Parkinson’s disease.

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
