# Peer review of "Has-miR-129-5p’s Involvement in Different Disorders, from Digestive Cancer to Neurodegenerative Diseases"

_biomedicines, 2023, doi:10.3390/biomedicines11072058_

Round 1

Reviewer 1 Report

A review is presented on the issues of regulation in the development of various human diseases. It is worth noting the importance of the identified factors obtained during the analysis of published works, as well as with the use of modern technologies. The conclusion fully reflects the conducted research. However, it is necessary to note the practical significance of the facts obtained.

A review is presented on the issues of regulation in the development of various human diseases. It is worth noting the importance of the identified factors obtained during the analysis of published works, as well as with the use of modern technologies. The conclusion fully reflects the conducted research. However, it is necessary to note the practical significance of the facts obtained.

Author Response

We thank the distinguish reviewer  for the time, encouragement and for the effort paid on our manuscript, we also added some figures to improve the clinical benefits outlined by our review.

Reviewer 2 Report

Dr. Adrian Boicean and their colleagues made an interesting systematic review entitled “Has- MIR-129-5 -p involved in different disorders, from 2 digestive cancer to neurodegenerative diseases”.

In this manuscript the authors elaborated the role of miRNA 129-5p involvement in 17 digestive cancer and neurological disorders.

In abstract: The authors should elaborate what is  microARN 129 5p mean?

Figure: 2 please improve the quality of the figure.

If possible the authors should include a elaborated figure mentioning the role of miRNA 129-5p in numerous cancers.

There are numerous typos throughout the manuscript, the authors should change accordingly.

Line: 348: WNT signaling is wrongly mentioned, please change.

I recommend the manuscript for acceptance after minor revisions.

can be improved

Author Response

In abstract: The authors should elaborate what is  microARN 129 5p mean?

We thank the distinguished reviewer for this suggestion, we added in the abstract more information about miR129-5p.

Figure: 2 please improve the quality of the figure. If possible the authors should include a elaborated figure mentioning the role of miRNA 129-5p in numerous cancers.

We appreciate the distinguished reviewers’ observation in this regard and have improved figure no 2 by mentioning the role of miR129-5p in various types of cancer.

There are numerous typos throughout the manuscript, the authors should change accordingly.

Line: 348: WNT signaling is wrongly mentioned, please change.

 We thank the distinguished reviewer for drawing our attention upon this error, we have corrected it.

We appreciate the distinguished reviewers’ observation in this regard and have corrected the formatting errors throughout the manuscript. English language was revised as well.

Reviewer 3 Report

This is an interesting review on miR-129-5p in digestive cancer and neurodegenerative diseases. Manuscript needs careful checking and proofreading. The description on WNT pathway may be revised to be more clear in terms of the involvement in digestive cancer and neurodegenerative diseases.

Figures need to be revised to be clear and comprehensive.

The careful editing is needed across the manuscript.

Specific Comments:

1. The study demonstrates that miR-129-5p plays a role in cancers and neurodegenerative diseases.
2. The review manuscript summarizes the roles of miR-129-5p in cancers including pancreatic cancer, hepatocarcinoma, and gastric cancer. The miR-129-5p seems to be involved in Wnt pathway. Since PI3K-Akt signaling pathway seems to be involved, some comprehensive figure to illustrate the role of miR-129-5p in cancer pathways.
3. Authors describe that miR-129-5p is involved in neurodegenerative disease. Section 3.2. may need some revisions to emphasize the perspective in neurodegenerative diseases and add a figure illustrating the role of miR-129-5p in the neurodegenerative diseases.
4. So many typos and inconsistency in the abbreviations are found in the manuscript. Please carefully proofread the manuscript.

Extensive editing is needed since a lot of typos and errors are included in the manuscript.

Author Response

This is an interesting review on miR-129-5p in digestive cancer and neurodegenerative diseases. Manuscript needs careful checking and proofreading. The description on WNT pathway may be revised to be more clear in terms of the involvement in digestive cancer and neurodegenerative diseases.

We appreciate the thoroughness of the distinguished reviewer’s evaluation. We have proofread the manuscript, and have corrected all formatting errors throughout it, we also revised and added more information about WNT pathway and its involvement in digestive cancer and neurodegenerative diseases.

Specific Comments:

  1. The study demonstrates that miR-129-5p plays a role in cancers and neurodegenerative diseases.

We thank the distinguished reviewer for the appreciation and provided suggestions.

  1. The review manuscript summarizes the roles of miR-129-5p in cancers including pancreatic cancer, hepatocarcinoma, and gastric cancer. The miR-129-5p seems to be involved in Wnt pathway. Since PI3K-Akt signaling pathway seems to be involved, some comprehensive figure to illustrate the role of miR-129-5p in cancer pathways.

We thank the distinguished reviewer for this observation. We revised figure 2 in order to highlight the implications of mir129 -5pin targeting different pathways, genes and proteins involved in digestive cancers.
3. Authors describe that miR-129-5p is involved in neurodegenerative disease. Section 3.2. may need some revisions to emphasize the perspective in neurodegenerative diseases and add a figure illustrating the role of miR-129-5p in the neurodegenerative diseases.

We highly appreciate the distinguished reviewer’s observation in this regard, we revised section nr 3 and added a figure illustrating the role of miR129 5p in neurodegenerative diseases.

  1. So many typos and inconsistency in the abbreviations are found in the manuscript. Please carefully proofread the manuscript.

We appreciate the thoroughness of the distinguished reviewer’s evaluation. We have proofread the manuscript, and have corrected all formatting errors throughout it.

Round 2

Reviewer 3 Report

The authors addressed the reviewer's comments.

Author Response

We thank the distinguished reviewer for the appreciation and support.